# Trends and Characteristics of Inter-Provincial Migrants in Mainland China and Its Relation with Economic Factors: A Panel Data Analysis from 2011 to 2016

**Lishuo Shi** [1,2,†]**, Wen Chen** [1,2,†]**, Jiaqi Xu** [1,2] **and Li Ling** [1,2,*]

[1]  Department of Medical Statistics, School of Public Health, Sun Yat-sen University, Guangzhou 510000, China; shilsh5@mail2.sysu.edu.cn (L.S.); chenw43@mail.sysu.edu.cn (W.C.); xujq35@mail2.sysu.edu.cn (J.X.)

[2]  Sun Yat-sen Centre for Migrant Health Policy, Sun Yat-sen University, Guangzhou 510000, China

\*  Correspondence: lingli@mail.sysu.edu.cn; Tel.: +86-020-8733-3319

†  Lishuo Shi and Wen Chen have equally contributed to the work and are co-first authors.

**Abstract:** For areas facing challenges associated with migration, information about the number of migrants and their demographic characteristics is needed to formulate regional development planning. This study analyzed the trends and characteristics of inter-provincial migrants in provinces in mainland China and related economic factors using panel regression models. The results showed that the number of inter-provincial migrants had increased in provincial municipalities, as had the proportions of female and elderly migrants. A higher annual net migration rate was associated with slower growth rate of real gross domestic product (RGDP) per capita and faster growth rates of the tertiary and secondary industry GDPs. The higher proportion of female migrants was related to the faster growth rate of the tertiary industry GDP and the lower proportion of the secondary industry in GDP. The proportion of youth migrants was positively related to educational investment, while the proportion of elderly migrants was positively related to financial expenditure per capita on culture and recreation. These empirical results were robust across different estimation methods, except the result about the proportion of elderly migrants. These findings further reveal the association between inter-provincial migration and economy and provide policy reference for the management of migrants.

**Keywords:** inter-provincial migrants; demographic characteristics; economic factors; mainland China

## 1. Introduction

As the world has entered "the age of migration" [1], the number of migrants reached 1 billion in 2015, including 740 million internal migrants worldwide [2]. Along with rapid economic development, many countries have experienced or are experiencing massive internal population mobility. For example, in the United States, a large number of people migrated to the rust belt of the Midwest before the 1970s [3]; in Japan, the population mainly migrated to the Tokyo, Osaka, and Nagoya areas, and these three areas accounted for 51.9% of the total population in Japan in 2016 [4]; in India, there were approximately 42 million inter-state migrants by 2011 [5]. As the country hosting the most internal migrants in the world, China's mass internal migration began in the late 1980s [6]. By the end of 2017, there were 245 million internal migrants in China [7]. Among them, there were 97 million inter-provincial migrants, mainly laborers [8], doubling in number since 2000 [9].

Internal migration is essentially a redistribution of the population. It plays a dominant role in determining the size, growth, and structure of the population in each area [10,11], which is particularly

important in the context of an aging population [12]. While the number of internal migrants is rising, the demographic structure of this population is changing. The gender imbalance of internal migrants has changed since the phenomenon of the "feminization of migration" emerged in the 1960s [13]. The proportion of women among the migrants in Europe and North America has increased to more than 50% [14], and this figure has reached 48.3% in China [7]. In terms of age structure, the youth labor force accounts for a dominant proportion of the internal migrants in many countries [2], while the proportions of the elderly and children have changed over time [15,16]. In China, the proportion of children under 18 among inter-provincial migrants decreased from 12.78% in 2005 to 10.95% in 2010, and the proportion of elderly over 60 increased from 2.64% in 2005 to 3.23% in 2015 [8,17].

With the rapid urbanization process in the past 30 years in China, some regions obtained preferential development policies and have vigorously promoted the construction of urbanization, and the distribution of social wealth has been further concentrated. Driven by economic factors, such as higher salary, a large number of rural residents are entering the city. However, under the household registration management policy in China, these individuals are defined as internal migrants and are classified as temporary residents in the city. In the cities, internal migrants were treated unequally in welfare resources allocation, which led to the fact that internal migrants were underprivileged groups [18]. The government has been aware of this problem and reformed the policies to improve the equity of resource allocation, such as the equalization of public health services [7]. In China, resource allocation and regional development planning are mostly made by provincial governments. Therefore, especially in areas with imbalanced inflows and outflows of migrants, a better understanding of the trends of the number and characteristics of inter-provincial migrant flows and driving factors may help policy makers plan for local development and the provision of social services, such as social welfare, education, and health services [19,20].

In the 20th century, several models of population migration were developed to study the laws of migration. Except for the gravity model [21], the Lewis two-sector model [22], Todaro model [23], and Everett push–pull model [24] all included regional economic differences as the fundamental driving force of population mobility. Empirical studies have found that the more economically developed provinces or states are, the higher the immigration rates are. This relationship has been verified in the United States, Japan [25], England [26], China [27,28], and countries in Sub-Saharan Africa [29]. Areas with developed economy can cause the migration of surplus rural laborers because of higher salaries [30,31].

Only a few studies have explored the association between the demographic structure of inter-provincial migrants and economic factors. Previous studies inferred that the social and economic situation in developed countries or provinces facilitates women's access to a variety of educational and employment opportunities [14,32] and may thereby attract more female migrants. Reports from the United Nations [14] and China [9] have shown that female migrants represent a larger proportion of the population in relatively developed areas than that in developing areas. Economic growth is usually accompanied by changes in the industrial structure, especially the development of the tertiary industry [33]. Thus, the growth of the tertiary industry may be an important factor in promoting female mobility, which leads to the change of the gender structure of the migrants. In terms of age structure, certain studies have found that the proportion of migrants with children in economically developed areas is significantly higher than that in economically developing areas [34,35], and whether children can obtain better education resources in the inflow areas affects adult migrant decisions of whether to take children with them [36]. Additionally, some studies have noted differences in the regional distribution of elderly migrants, but the causes of these differences have not been well explored [16,37]. Previous studies indicated elderly migrants have great need for health care [38] and cultural and recreational activities [39]. In general, the economic factors regarding the education, health, culture, and entertainment of the province may have an impact on the age structure of inter-provincial migrants.

In summary, although existing studies have indicated that economic factors might be related to the number and the gender–age structure of inter-provincial migrants, few studies have verified this

relationship by using continuous dynamic data at the macro level in China. Most of the existing studies are cross-sectional studies, which cannot eliminate the confounding effects of the unique geographical environment and climate of each region on migration [40]. In addition, most previous studies have used a single economic indicator, such as GDP per capita [29,41] or income per capita [25,31], to explore the relations between the economy and migration. Since subgroups of migrants, such as children and the elderly, have different needs for social resources, the increasing number of migrants and population aging raise the need for further analyses of the relationship between comprehensive economic indicators and the number and structure of migrants. Such studies may better understand the migration of subgroups to facilitate the provision and better management of the necessary services for migrants.

Therefore, the purpose of this study is (1) to describe the changing trend of inter-provincial migrants in China from 2011 to 2016 and (2) to analyze the relationship between economic factors and the number and gender–age structure of inter-provincial migrants in mainland China at the macro level.

## 2. Materials and Methods

### 2.1. Data Sources

The data used in this study were from the following three databases:

1. The 6th national population census of China, which covers all 31 provincial-level jurisdictions in mainland China. In each province, the census data include the number of inter-provincial immigrants, number of emigrants, population with household registration, and resident population. According to China's household registration management system [42], an individuals' place of origin is his or her place of household registration. People who leave the place of household registration and move to other provinces for more than six months are defined as inter-provincial emigrants. In the host provinces, inter-provincial migrants are classified as temporary residents rather than household registered residents. Therefore, the resident population of a province includes temporary residents (i.e., inter-provincial immigrants) and household registered residents;

2. China Statistical Yearbook 2012–2017, which provides information on "number of resident population", "growth rate of population", "natural growth rate of population", "Real Gross Domestic Product (RGDP)", "growth rate of tertiary industry GDP", " growth rate of secondary industry GDP", "the proportion of tertiary industry in GDP", "the proportion of secondary industry in GDP", "urbanization rate", "educational investment", "education financial expenditure", "cultural and recreational financial expenditure", and "health financial expenditure" of provinces from 2011 to 2016;

3. China National Dynamic Monitoring Data of Internal Migrants from 2011 to 2016, which were collected by the National Health Commission of the People's Republic of China (www.http://chinaldrk.org.cn). It is the most comprehensive and authoritative survey in mainland China, with representative samples of the internal migrants in all 31 provincial-level jurisdictions. The participants in the survey from 2011–2014 and 2015–2016 were internal migrants aged 15–59 and aged 15 and over, respectively, who had lived in the study area for more than one month. This survey collected information concerning the place of origin, current residence, gender, and age of the respondents and their family members living with them, including children and the elderly.

### 2.2. Construction of the Indices or Variables

#### 2.2.1. Descriptive Indices of Inter-Provincial Migration

RNMi (Revised Net Migration Rate) and RGMi (Revised Gross Migration Rate) (Based on the First Database)

The net migration rate (*NM*) and the gross migration rate (*GM*) are indexes often used in previous research to evaluate the migration of population in regions [29,43]. The expressions are as follows:

$$NM_i = \frac{I_i - O_i}{P_i}, \tag{1}$$

$$GM_i = \frac{I_i + O_i}{P_i}, \tag{2}$$

where $I_i$ is the number of immigrants of province $i$, $O_i$ is the number of emigrants of province $i$, and $P_i$ is the resident population of province $i$. However, this method did not take the share of the migrants of one region to the national total migrants into account, which might result in overestimation or underestimation errors to both indices considering the huge difference of population size in provinces in China. For example, in regions with a small number of immigrants and total population but a relatively high proportion of immigrants to total population, usually in the western provinces with low population density such as Tibet, the two indices may be overestimated; while on contrary, in regions with a large number of immigrants and total population but a relatively small proportion of immigrants to total population, usually in provinces in east China such as Fujian, the two indices may be underestimated. To alleviate this problem, based on the identification method of regional types proposed by Liu [43], *RNM* and *RGM* were adopted in this study to describe the migration in each province. The two indexes considered both the relative number and absolute number of migrants. The expressions are as follows:

$$RNM_i = \begin{cases} NM_i * \frac{I_i}{\sum_{i=1}^{n} I_i} * n \ (if \ NM_i > 0) \\ NM_i * \frac{O_i}{\sum_{i=1}^{n} O_i} * n \ (if \ NM_i < 0) \end{cases}, \tag{3}$$

$$RGM_i = GM_i * \frac{I_i + O_i}{\sum_{i=1}^{n} I_i + \sum_{i=1}^{n} O_i} * n, \tag{4}$$

$$Ra = a * \overline{RGM_i}, \ \overline{RGM_i} = \frac{\sum_{i=1}^{n} RGM_i}{n}, \tag{5}$$

where $n$ is the total number of provinces in mainland China, and $\overline{RGM_i}$ is the national average gross migration rate for all provinces. If the migrants in each province were evenly distributed, then the *RNM* = *NM*; if all the inter-provincial migrants flowed to province $i$, then the $RNM_i = NM_i*n$. Thus, the *RNM* could be more than 100%, and a higher *RNM* indicated more frequent population migration in the province. *a* is the threshold value of $GM_i$ for distinguishing the provinces with active migration from those without, which previous studies usually set at 10% [43,44]. However, the population size of China is enormous. Although the index of some provinces does not exceed 10%, the millions of migrants in these provinces will still have significant impacts on many aspects of the public field, so these provinces are unreasonable to be classified as inactive areas. Thus, we weighted the threshold with the national average gross migration rate to identify the provinces with relatively active migration in China, and *Ra* is the modified index of threshold values.

Based on $RNM_i$, $RGM_i$, and $Ra$, four different types of provinces are identified:

- Type 1—Active net-immigration province, when $RGM_i > Ra$ and $RNM_i > Ra$;
- Type 2—Active balanced-migration province, when $RGM_i > Ra$ and $-Ra \leq RNM_i \leq Ra$;
- Type 3—Active net-emigration province, when $RGM_i > Ra$ and $RNM_i < -Ra$; and

- Type 4—Inactive migration province, when $RGM_i \leq Ra$.

The threshold $Ra$ in this study was calculated to be 3%.

### 2.2.2. Distribution of Migration (Based on the Third Database)

This index is used to measure the proportion of immigration to each province that comes from a specific sending province. This is one of the components of the migration framework outlined by Willekens and Baydar [45] and is often used to analyze trends in migration [46].

### 2.2.3. Independent Variables and Economic Variables (Based on the Second Database)

#### The Growth Rate of Real GDP (RGDP) per Capita (GRR)

The growth rate of RGDP per capita is widely used as an indicator of economic development [47] and has been proven to be related to migration flows [48]. The GDP is the sum of the gross value added by all resident producers in the economy plus any product taxes and minus any subsidies not included in the value of the products. RGDP per capita is the GDP adjusted for inflation and then divided by the resident population.

#### The Growth Rate of Tertiary Industry GDP (GRT) and Secondary Industry GDP (GRS)

According to modern economic theory, the tertiary industry mainly includes the Internet, financial, and service industries [33], while the secondary industry mainly includes manufacturing and construction industries. The two indicators are commonly used to represent economic development in ecological research [49]. The imbalance of industrial development among provinces is an essential reason for economic disparity, so these two factors may also be related to migration flows.

#### The Proportion of Tertiary Industry in GDP (PTI) and Secondary Industry in GDP (PSI)

These two indicators reflect the regional industrial structure and are widely used in studies [50,51]. Generally, the proportion of the tertiary industry is positively correlated with regional economic development. Different industries are suitable for migrants of different genders and ages [52], so the adjustment of the industrial structure may be related to the gender and age structure of migrants in a province.

#### Urbanization Rate (UR)

The urbanization rate is the proportion of people living in urban areas to the total resident population of the province [44]. People in rural areas leave their hometown for towns in which industrial jobs are more widely available and attractive. This process typically goes hand-in-hand with rapid economic development and rising incomes. Thus, the urbanization rate is closely related to population mobility and economic development [53].

#### Educational Investment (EI) (Hundred Million Yuan)

Educational investment is an economic index used to evaluate the development of education resources [54]. The more a province invests in education, the better educational resources the younger generation will have, which might attract more families with educational needs.

#### Financial Expenditure per Capita on Education (FEE), Culture and Recreation (FEC), and Health (FEH) (Ten Thousand Yuan)

The elderly in China migrate to other provinces mainly to take care of their grandchildren or pursue a better retirement life [55]. This study selected these three economic variables related to the reasons to explore their effects on the migration of the elderly.

### 2.2.4. Dependent Variables

Annual Net Migration Rates per Thousand Population (ANM) (Based on the Second Database)

The *ANM* was the most intuitive indicator of population size change caused by population mobility [56]. It is calculated by subtracting the natural growth rate of the population from the growth rate of population in this study. This indicator represents the permillage of the net increase of population by migration to the total population in the province in the year.

The Sex Ratio of Migrants (SR) (Based on the Third Database)

The sex ratio refers to the ratio of male to female migrants. It is an important indicator of population structure.

Youth Dependency Ratio (YDR) and Elderly Dependency Ratio (EDR) of Migrants (Based on the Third Database)

The *YDR* and *EDR* are commonly used as indicators of population age structure [57], and they refer to the proportions of youth and the elderly relative to laborers, respectively. According to the age structure division standard of Edward Rossett [58] and considering the current retirement age in China [59], we chose the age under 14 (including 14) as the standard for youth and the age over 60 (including 60) as the standard for the elderly. Their expressions are as follows:

$$YDR = \frac{N_{0-14}}{N_{15-59}} * 100\%, \tag{6}$$

$$EDR = \frac{N_{60\sim}}{N_{15-59}} * 100\%, \tag{7}$$

where $N_{0-14}$ is the number of migrants between 0 and 14 years old in each province in the database, $N_{15-59}$ is the number of migrants between 15 and 59 years old, and $N_{60\sim}$ is the number of migrants 60 years old and older.

### 2.3. Empirical Methodologies

First, we calculated $RNM_i$, $RGM_i$, and *Ra* to classify provinces into different types. Then, we described the trend of net inflow migrants and the sending provinces in each active net-immigration province from 2011 to 2016. Because the active net-immigration provinces absorbed the majority of the inter-provincial migrants in China [60], the labor force and the demand for social resources in these provinces are much more affected by inter-provincial migrants than those in other types of provinces. Then, we took the *ANM*, *SR*, *YDR*, and *EDR* of the active net-immigration provinces as dependent variables to build panel data models for further analyses. The panel data model was adopted in this study because it can account for the variations of variables over time and across provinces and comprehensively analyze the effects of economic variables on dependent variables [61]. First, we selected the economic variables that may influence the dependent variables through a literature review. Then, to avoid spurious regression of the panel data model, we used a panel unit root test to ensure the variables in one model were stationary or integrated with the same order. The Pedroni panel cointegration test was used to prove the existence of a long-term relationship between the variables integrated with the same order, and at least four out of the seven test statistics of the Pedroni test statistics were significant [61]. Since the China National Dynamic Monitoring Data of Internal Migrants first took the elderly over 60 years old as the survey participants in 2015, the *EDR* could be accurate only in 2015 and 2016. Thus, the time series length of the *EDR* and its related independent variables were too short to require stationarity and cointegration tests, so these variables were used directly to construct a panel regression model.

Next, we used Eviews 9.0 to estimate the model parameters and obtain the final results. We established panel data regression models by the ordinary least squares (OLS) method as follows:

$$y_{it} = \alpha_0 + \alpha_i + \theta_i\, t\ + x_{it}\beta + \varepsilon_{it}, \tag{8}$$

where *i* represents the province, *t* represents the year, *y* represents the dependent variables, *x* represents the independent variables, *β* represents the influence coefficient of the independent variables on the dependent variables, $\alpha_0$ represents the intercept, and $\varepsilon_{it}$ represents errors. The coefficients $\alpha_i$ (individual intercept) and $\theta_i$ (individual trend) are member-specific, vary across each cross-section, and define the deterministic trend specification. The Hausman test was used to test the appropriateness of the random effect of the cross-section and deterministic trend [62]. If the *p* value of the Hausman test was greater than 0.1, it indicated that the individual intercept and individual trend were not related to the explanatory variables, and the random effect model was applicable; if the *p* value of the Hausman test was less than 0.1, it indicated that the individual intercept and individual trend were related to the explanatory variables, and the fixed effect model was applicable. We chose the white diagonal as the coefficient covariance method to reduce the influence of heteroscedasticity of variables on parameters [63].

However, OLS results may be biased due to omitted variables or simultaneity. For example, although the country fixed effect (individual intercept) has been used to control for unobserved time-invariant factors, there might still be some "time-variant" variables that are omitted. There is also likely to be a simultaneity problem if the number and the demographic characteristics of migrants also affect economic variables of the provinces. In both cases, error terms will be correlated with the target explanatory variable, and the OLS estimation will be biased. To deal with this problem, we could utilize the two-stage least-squares (2SLS) method to analyze the panel model constructed by stationary variables, and we could utilize the dynamic OLS (DOLS) and fully modified OLS (FMOLS) methods to analyze the panel model constructed by variables integrated with the same order.

Finding the instrumental variable *Z* is crucial to build the 2SLS model. To summarize, the two conditions for a valid instrumental variable are exogenous, namely Cov (Z, error term) = 0, and relevance: Cov (Z, independent variable) ≠ 0; or as Levitt has mentioned, *Z* should affect the dependent variable only through independent variables [64]. Construction of the 2SLS model is divided into two steps. First, to obtain the instrumental variable estimation, the independent variable on instrumental variable *Z* and all other regressors are regressed at the first stage. The equation is as follows:

$$d_{it} = \alpha'_0 + \alpha'_i + \theta'_i\, t\ + Z_{it}\delta + x_{it}\beta' + \varepsilon'_{it}, \tag{9}$$

where *d* represents the independent variable, *Z* represents the instrumental variable, *x* represents the covariates, $\varepsilon'_{it}$ represents errors, and $\alpha'_0$, $\alpha'_i$, $\theta'_i$, $\delta$, and $\beta'$ represent the coefficients to be estimated similar to OLS. This obtains the predicted value of the independent variable, namely $\hat{d}_{it}$, that is then entered into the second stage to obtain the unbiased estimator of independent variable. The equation is as follows:

$$y_{it} = \alpha_0 + \alpha_i + \theta_i\, t\ + \hat{d}_{it}\gamma + x_{it}\beta + \varepsilon_{it}, \tag{10}$$

where *γ* represents the unbiased correlation coefficient of the independent variable that we focused on the dependent variable, and *x* represents the covariates. Other parameters have the same meaning as Equation (8). According to Staiger and Stock [65], if the instrument is weak, then normal distribution provides a poor approximation to the sampling distribution of the 2SLS. This bias is correlated with the first stage *F*-test value of the null hypothesis that the coefficient of the instrument (*Z*) equals to zero. They argue the "rule of thumb" for weak instrumental variable tests should be larger than 10 [66].

DOLS and FMOLS were the two most commonly used methods to analyze panel models constructed by variables integrated with the same order. The FMOLS estimator used a nonparametric method to modify the OLS estimator, while the DOLS estimator used the past and future values of the independent variable [67,68]. Both of them can effectively reduce the influence of the problem of endogeneity on the estimated parameters, and DOLS estimator outperforms the FMOLS method,

particularly with small samples [69]. For these reasons, we used these two methods to estimate the long-term relationship between variables integrated with the same order.

## 3. Results

### 3.1. Descriptive Statistics of Inter-Provincial Migration

According to the distribution of migrants in 2010, there were nine active net-immigration provinces, nine active net-emigration provinces, 12 active balanced-migration provinces, and one inactive migration province. The active net-immigration provinces attracted 78.72% of the inter-provincial migrants. Therefore, the characteristics of inter-provincial migrants in these nine provinces can well represent the inter-provincial migrants in mainland China. The inter-provincial migrants mostly flowed from the central provinces to the eastern coastal provinces. According to the revised gross migration rate (*RNM*) of the provinces, Guangdong, Shanghai, and Beijing were the three provinces with the largest migrant inflow, and Guizhou, Anhui, and Sichuan were the three provinces with the largest migrant outflow. See Table 1 and Figure 1.

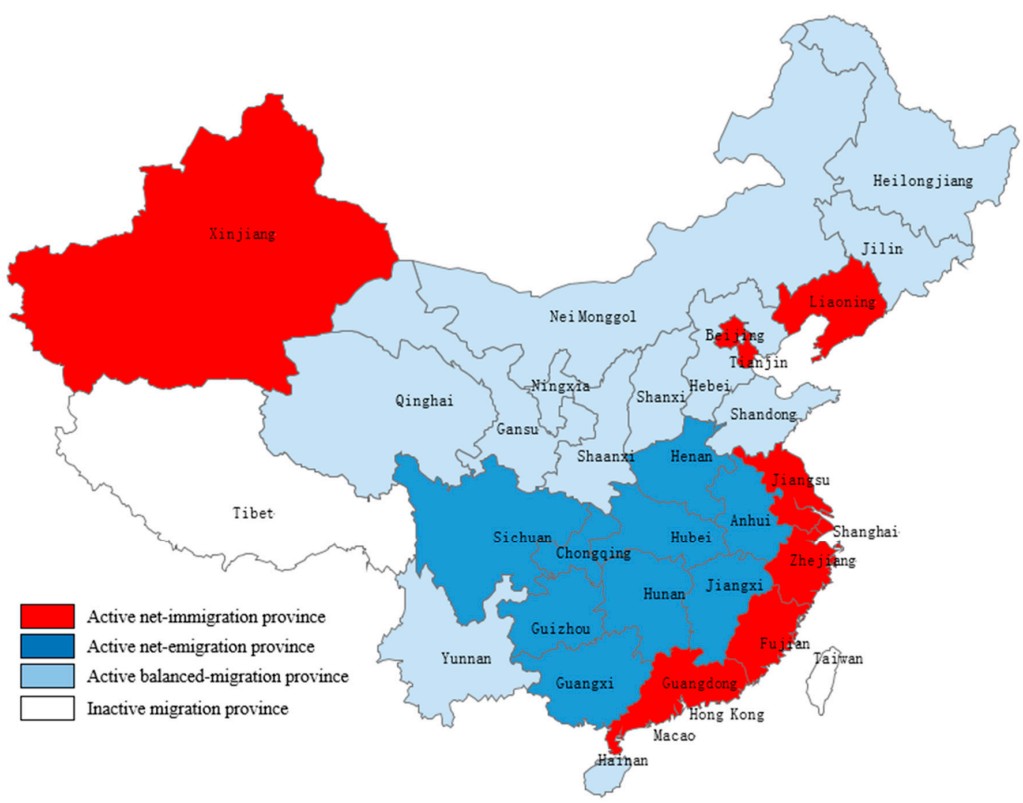

**Figure 1.** Provincial types of migrants in mainland China in 2010. Note: Information about Hong Kong, Macao, and Taiwan is not included.

The trends in the inter-provincial number of migrants from 2011 to 2016 showed that the active net-immigration provinces maintained a large number of migrant inflows (Table 2). The number of inter-provincial migrants in Beijing, Tianjin, and Shanghai still had significant growth, but the annual net migration rate (*ANM*) of each were decreasing, while the *ANM* in Xinjiang and Guangdong provinces were accelerating. The *ANM* of the other active net-immigration provinces were close to zero, which indicated the numbers of inter-provincial migrants in these provinces had little change. In Beijing, Tianjin, and Shanghai provinces, the increase of population mainly was due to the inflow of migrants. In Zhejiang and Fujian provinces, population growth had little to do with population mobility, while in Xinjiang and Guangdong, population growth was due to the dual effects of natural population growth and population mobility.

**Table 1.** Provincial types of migrants in mainland China based on the 2010 Census Data.

| Type of Province | Indices | Number of Provinces | Immigrants ($10^4$) (%) | Emigrants ($10^4$) (%) | Net Migrants ($10^4$) | Provincial Distribution (*RNM*, %) |
|---|---|---|---|---|---|---|
| Active net-immigration province | *RGM* > 3% *RNM* > 3% | 9 | 6760.58 (78.72) | 1431.04 (15.76) | 5329.54 | Guangdong (114.88), Shanghai (61.59), Beijing (44.98), Zhejiang (38.81), Jiangsu (15.35), Tianjin (12.41), Fujian (10.03), Xinjiang (5.37), Liaoning (4.34) |
| Active net-emigration province | *RGM* > 3% *RNM* < −3% | 9 | 732.70 (8.53) | 6515.32 (71.75) | −5782.62 | Guizhou (−33.32), Anhui (−33.24), Sichuan (−32.26), Henan (−26.51), Guangxi (−16.81), Chongqing (−14.89), Hunan (−9.28), Hubei (−8.82), Jiangxi (−3.13) |
| Active balanced-migration province | *RGM* > 3% −3% ≤ *RNM* ≤ 3% | 12 | 1077.81 (12.55) | 1128.23 (12.42) | −50.42 | Shandong (2.65), Shanxi (2.62), Yunnan (2.12), Nei Monggol (1.78), Hainan (0.99), Hebei (0.97), Jilin (0.73), Qinghai (0.61), Heilongjiang (0.37), Ningxia (0.29), Shaanxi (−0.84), Gansu (−2.89) |
| Inactive migration province | *RGM* ≤ 3% | 1 | 16.54 (0.19) | 5.72 (0.06) | 10.82 | Tibet (0.28) |

Note: *RGM*, revised gross migration rate; *RNM*, revised net migration rate. Part of the emigrants went abroad, so the number of total emigrants was larger than that of the total immigrants.

**Table 2.** Growth rate of population and annual net migration rates of provinces in active net-immigration provinces from 2011 to 2016.

| Active Net-Immigration Province | 2011 | | 2012 | | 2013 | | 2014 | | 2015 | | 2016 | |
|---|---|---|---|---|---|---|---|---|---|---|---|---|
| | *GR* (‰) | *ANM* (‰) | *GR* (‰) | *ANM* (‰) | *GR* (‰) | *ANM* (‰) | *GR* (‰) | *ANM* (‰) | *GR* (‰) | *ANM* (‰) | *GR* (‰) | *ANM* (‰) |
| Beijing | 29.05 | 25.03 | 24.76 | 20.02 | 22.23 | 17.82 | 17.49 | 12.66 | 8.83 | 5.82 | 0.92 | −3.20 |
| Tianjin | 43.11 | 40.61 | 42.80 | 40.17 | 41.75 | 39.47 | 30.57 | 28.43 | 19.77 | 19.55 | 9.70 | 7.87 |
| Liaoning | 1.83 | 2.17 | 1.36 | 1.76 | 0.23 | 0.26 | 0.23 | −0.03 | −2.05 | −1.63 | −0.91 | −0.73 |
| Shanghai | 19.10 | 17.23 | 14.06 | 9.86 | 14.70 | 11.76 | 4.55 | 1.41 | −4.53 | −6.98 | 2.07 | −1.93 |
| Jiangsu | 3.81 | 1.20 | 2.66 | 0.21 | 2.40 | −0.03 | 2.64 | 0.22 | 2.01 | −0.01 | 2.88 | 0.15 |
| Zhejiang | 2.94 | −1.13 | 2.56 | −2.04 | 3.83 | −0.72 | 1.82 | −3.18 | 5.63 | 0.61 | 9.21 | 3.51 |
| Fujian | 7.31 | 1.10 | 7.53 | 0.52 | 6.94 | 0.75 | 8.48 | 0.98 | 8.67 | 0.87 | 9.12 | 0.82 |
| Guangdong | 6.13 | 0.03 | 8.47 | 1.52 | 4.72 | −1.30 | 7.51 | 1.42 | 11.66 | 4.86 | 13.83 | 6.39 |
| Xinjiang | 10.98 | 0.41 | 10.86 | 0.02 | 13.88 | 2.96 | 15.02 | 3.55 | 26.98 | 15.90 | 16.10 | 5.02 |

Note: *GR*, growth rate of population; *ANM*, annual net migration rate.

The sending provinces of immigrants in different active net-immigration provinces differed widely in geographical location and underwent no significant changes from 2011 to 2016. The inter-provincial migrants in Liaoning province mainly came from northeast China, while that in Fujian and Zhejiang provinces mainly came from east and southwest China, that in Beijing and Tianjin mainly came from north and east China, that in Jiangsu and Shanghai mainly came from east China, and that in Guangdong mainly came from central and south China. Except Xinjiang, most of the inter-provincial migrants in active net-immigration provinces came from the same or neighboring regions. In several active net-immigration provinces (Tianjin, Jiangsu, Beijing, and Guangdong), the proportions of inter-provincial migrants from neighboring provinces (Shandong, Anhui, Hebei, and Guangxi, respectively) obviously increased over the period. See Figure 2. In addition, the proportion of immigrants from western provinces, such as Chongqing and Sichuan, declined in active net-immigration provinces.

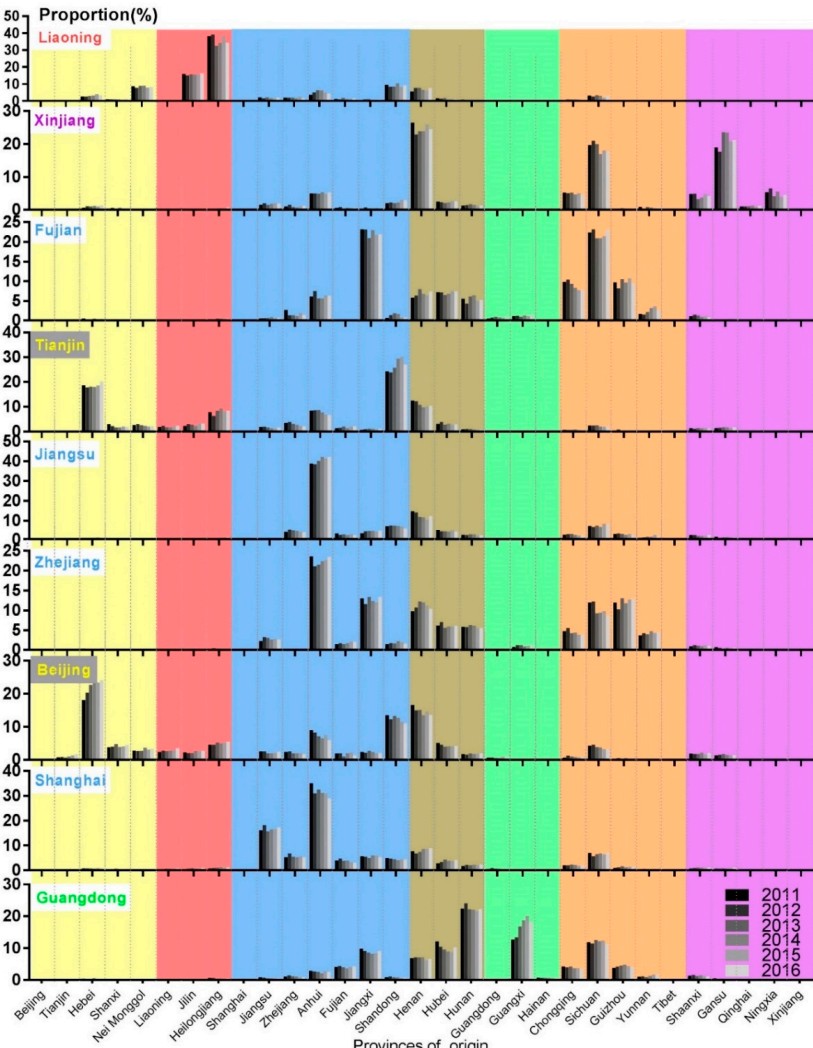

**Figure 2.** Distribution and trends of sending provinces of inter-provincial migrants in active net-immigration provinces from 2011 to 2016. Note: yellow region represents North China: Beijing, Tianjin, Hebei, Shanxi, and Nei Monggol; red region represents Northeast China: Liaoning, Jilin, and Heilongjiang; blue region represents East China: Shanghai, Jiangsu, Zhejiang, Anhui, Fujian, Jiangxi, and Shandong; brown region represents Central China: Henan, Hubei, and Hunan; green region represents South China: Guangdong, Guangxi, and Hainan; orange region represents Southwest China: Chongqing, Sichuan, Guizhou, Yunnan, and Tibet; purple region represents Northwest China: Shaanxi, Gansu, Qinghai, Ningxia, and Xinjiang.

The age and gender structure of inter-provincial migrants showed distinct differences among active net-immigration provinces. The youth dependency ratios (*YDRs*) were higher in Tianjin, Xinjiang, and Guangdong than in other provinces. The changing *YDRs* of Beijing and Liaoning showed a decreasing trend, while that of Guangdong and Jiangsu showed an increasing trend. The changing *YDRs* of the other provinces showed overall consistent trends. Beijing, Shanghai, and Liaoning had higher elderly dependency ratios (*EDRs*) than other provinces. The *EDRs* of many provinces increased, except for Zhejiang and Shanghai provinces. The sex ratio was lower in Beijing and Shanghai than in other provinces and showed a decreasing trend, while the sex ratios of other provinces fluctuated. See Table 3.

**Table 3.** Age and gender structure of inter-provincial migrants in active net-immigration provinces from 2011 to 2016.

| Province | 2011 | 2012 | 2013 | 2014 | 2015 | 2016 |
|---|---|---|---|---|---|---|
| **Liaoning** | N = 5240 | N = 5610 | N = 7067 | N = 7133 | N = 6924 | N = 6670 |
| YDR (%) | 21.88 | 20.82 | 23.24 | 22.79 | 15.76 | 17.52 |
| EDR (%) | – | – | – | – | 7.10 | 7.10 |
| Sex ratio | 112.04 | 115.42 | 113.26 | 121.58 | 108.55 | 108.94 |
| **Xinjiang** | N = 8103 | N = 8256 | N = 7372 | N = 7402 | N = 11704 | N = 9207 |
| YDR (%) | 27.96 | 27.75 | 26.01 | 27.59 | 31.21 | 28.47 |
| EDR (%) | – | – | – | – | 3.12 | 3.32 |
| Sex ratio | 112.9 | 113.22 | 119.78 | 116.73 | 110 | 111.24 |
| **Fujian** | N = 5307 | N = 5411 | N = 9937 | N = 8921 | N = 9453 | N = 9727 |
| YDR (%) | 22.32 | 21.74 | 24.37 | 24.21 | 22.68 | 24.80 |
| EDR (%) | – | – | – | – | 1.14 | 1.39 |
| Sex ratio | 115.05 | 122.87 | 119.63 | 118.77 | 109.95 | 113.17 |
| **Tianjin** | N = 9093 | N = 9890 | N = 14811 | N = 15742 | N = 15534 | N = 12328 |
| YDR (%) | 26.05 | 26.60 | 26.92 | 29.15 | 27.81 | 26.43 |
| EDR (%) | – | – | – | – | 2.00 | 2.48 |
| Sex ratio | 116.78 | 114.41 | 111.15 | 117.06 | 112 | 106.65 |
| **Jiangsu** | N = 9013 | N = 12665 | N = 19742 | N = 19653 | N = 19209 | N = 13534 |
| YDR (%) | 18.76 | 22.02 | 24.42 | 24.00 | 23.31 | 24.18 |
| EDR (%) | – | – | – | – | 2.08 | 2.72 |
| Sex ratio | 106.61 | 113.9 | 112.54 | 114 | 111.91 | 112.63 |
| **Zhejiang** | N = 11380 | N = 21412 | N = 29500 | N = 29949 | N = 29827 | N = 21529 |
| YDR (%) | 19.66 | 21.00 | 21.54 | 21.73 | 21.24 | 20.06 |
| EDR (%) | – | – | – | – | 1.40 | 1.21 |
| Sex ratio | 110.13 | 115.89 | 113.04 | 115.84 | 110.66 | 114.13 |
| **Beijing** | N = 9375 | N = 14740 | N = 18586 | N = 18617 | N = 20151 | N = 16630 |
| YDR (%) | 26.52 | 24.89 | 24.25 | 24.40 | 24.80 | 21.74 |
| EDR (%) | – | – | – | – | 5.20 | 7.43 |
| Sex ratio | 110.22 | 107.04 | 108.51 | 110.79 | 103.38 | 100.4 |
| **Shanghai** | N = 10156 | N = 36870 | N = 20139 | N = 19859 | N = 20789 | N = 17834 |
| YDR (%) | 21.89 | 20.83 | 23.24 | 22.79 | 21.70 | 22.04 |
| EDR (%) | – | – | – | – | 4.87 | 4.28 |
| Sex ratio | 107.73 | 107.17 | 108.64 | 108.68 | 106.4 | 99.42 |
| **Guangdong** | N = 16055 | N = 19278 | N = 19341 | N = 20387 | N = 25915 | N = 16767 |
| YDR (%) | 22.79 | 23.08 | 25.92 | 27.93 | 29.31 | 28.93 |
| EDR (%) | – | – | – | – | 1.52 | 1.61 |
| Sex ratio | 105.13 | 110.08 | 109.82 | 111.5 | 107.47 | 107.12 |

Note: N, number of inter-provincial migrants whose demographic information was collected; *YDR*, youth dependency ratio; *EDR*, elderly dependency ratio; sex ratio, female = 100; data are missing because there is no information on migrants over 60 years old in these years.

*3.2. Results of the Panel Data Model*

3.2.1. Descriptive Statistics, Panel Unit Root Tests for Variables, and Pedroni Tests for Panel Cointegration

　　　Descriptive results showed that in these active net-immigration provinces the average annual net migration rate *(ANM)* was 6.148%, there were more males than females, the average youth dependency ratio (*YDR*) of inter-provincial migrants was 23.946%, while the average elderly dependency ratio (*EDR*) was 3.331%. *ANM* and *EDR* had higher variance, while the sex ratio (*SR*) and the youth dependency ratio (*YDR*) had lower variance. The results of the panel unit root tests showed that seven variables, namely, annual net migration rate (*ANM*), sex ratio (*SR*), growth rate of RGDP per capita (*GRR*), growth rate of tertiary industry GDP (*GRT*), growth rate of secondary industry GDP (*GRS*), urbanization rate (*UR*), and proportion of secondary industry in GDP (*PSI*) were stationary at the 5% significance level, while the other three variables, namely, *YDR*, educational investment (*EI*), and the proportion of tertiary industry in GDP (*PTI*), were stationary in the first difference. The Levin, Lin, and Chu tests and the Augmented Dickey-Fuller (ADF)-Fisher chi-square tests yielded the same results (Table 4). The results of the cointegration test showed evidence for the existence of a long-term relationship between the youth dependency ratio (*YDR*), educational investment (*EI*), and the proportion of tertiary industry in GDP (*PTI*) by four out of the seven Pedroni test statistics (Table 5).

**Table 4.** Results of the descriptive statistics and panel unit root test for variables.

| Variables | Mean (SD) | Observations | Levin, Lin, and Chu *t* | | ADF-Fisher Chi-Square | |
| --- | --- | --- | --- | --- | --- | --- |
| | | | Level | First Difference | Level | First Difference |
| **Dependent** | | | | | | |
| ANM (%) | 6.148 (11.146) | 54 | −5.43(0.00) | – | 49.22(0.00) | – |
| SR (%) | 111.517 (4.771) | 54 | −10.48(0.00) | – | 32.48(0.02) | – |
| YDR (%) | 23.946 (3.158) | 54 | 0.63(0.74) | −5.15(0.00) | 10.56(0.91) | 46.55(0.00) |
| EDR (%) | 3.331 (2.163) | 18 | – | – | – | – |
| **Independent** | | | | | | |
| GRR (%) | 7.499 (2.393) | 54 | −48.90(0.00) | – | 33.45(0.01) | – |
| GRT (%) | 9.535 (1.953) | 54 | −14.01(0.00) | – | 28.98(0.02) | – |
| GRS (%) | 8.289 (4.385) | 54 | −8.11(0.00) | – | 57.49(0.00) | – |
| UR (%) | 69.712 (13.173) | 54 | −2.93(0.00) | – | 32.46(0.01) | – |
| EI | 216.175 (138.498) | 54 | 9.96(1.00) | −3.31(0.00) | 4.67(0.99) | 33.35(0.01) |
| PTI (%) | 50.829 (12.285) | 54 | 3.45(0.99) | −8.46(0.00) | 2.30(1.00) | 39.50(0.00) |
| PSI (%) | 43.405 (9.590) | 54 | −11.29(0.00) | – | 98.10(0.00) | – |
| FEE | 0.259 (0.079) | 18 | – | – | – | – |
| FEC | 0.035 (0.021) | 18 | – | – | – | – |
| FEH | 0.109 (0.033) | 18 | – | – | – | – |

Note: ANM, annual net migration rate; SR, sex ratio, female = 100; YDR, youth dependency ratio; EDR, elderly dependency ratio; GRR, growth rate of RGDP per capita; GRT, growth rate of tertiary industry GDP; GRS, growth rate of secondary industry GDP; UR, urbanization rate; PSI, the proportion of secondary industry in GDP; PTI, the proportion of tertiary industry in GDP; EI, educational investment; FEE, financial expenditure per capita on education; FEC, financial expenditure per capita on culture and recreation; FEH, financial expenditure per capita on health. ADF, Augmented Dickey-Fuller. The null hypothesis for the two panel unit root tests corresponds to the presence of unit root (non-stationary). *P*-values are reported in parentheses. The lag selection is based on the automatic Akaike Information Criteria (AIC).

**Table 5.** Result of Pedroni panel cointegration tests.

| Indexes | Statistic | *P* Value |
|---------|-----------|-----------|
| Panel v-Statistic | −1.55 | 0.94 |
| Panel rho-Statistic | 1.36 | 0.91 |
| Panel PP-Statistic | −5.99 | 0.00 |
| Panel ADF-Statistic | −8.48 | 0.00 |
| Group rho-Statistic | 2.91 | 0.99 |
| Group PP-Statistic | −5.98 | 0.00 |
| Group ADF-Statistic | −15.01 | 0.00 |

Note: The null hypothesis is no cointegration against the alternative of the presence of cointegration.

### 3.2.2. Panel Data Regression Models

Based on the results of the unit root test and cointegration test and a comprehensive literature review, we selected independent variables for four dependent variables. The results are shown in Table 6. The annual net migration rate *(ANM)* and economic growth indicators including growth rate of RGDP per capita (*GRR*), growth rate of tertiary industry GDP (*GRT*), and growth rate of secondary industry GDP (*GRS*) were regressed in Model 1, Model 2, and Model 3. We focused on the relationship between the growth rate of RGDP per capita and the annual net migration rate, while the growth rate of tertiary industry GDP and the growth rate of secondary industry GDP were put in models as covariates. Theoretically, the change of urbanization rate (*UR*) in provinces mainly was due to the transfer of migrants from rural to urban within the province, which should not have a direct relation to the annual net migration rate of inter-provincial migrants. Data analysis also indicated that the direct correlation coefficient between urbanization rate and the annual net migration rate was not statistically significant (not shown in the table), while the urbanization rate was closely related to the growth rate of RGDP per capita (coefficient = −0.520, P < 0.001). Thus, the urbanization rate was used to instrument the growth rate of RGDP per capita in the first-stage in Model 2, while the urbanization rate and the growth rate of RGDP per capita lagged by one year and were used to instrument the growth rate of RGDP per capita in Model 3. The *F* value of the first stage in Model 2 was larger than 10, which indicated the urbanization rate was not a weak instrument variable. The 2SLS model (Model 2) found that a higher annual net migration rate was associated with a slower growth rate of RGDP per capita (coefficient = −7.829, P < 0.001), which was consistent with the results of the OLS model (Model 1) (coefficient = −6.459, P < 0.001) and Model 3 (coefficient = −6.442, P < 0.001), indicating that the negative correlation was robust. This result meant that the growth rate of RGDP per capita increased by 1 percentage point for every 7.829 permillage points of the annual net migration rate decrease. Also, we found the growth rate of tertiary industry GDP and the growth rate of secondary industry GDP were positively correlated with the annual net migration rate (coefficient = 4.64 and 3.88, respectively, P < 0.001) in Model 2. Even the coefficients may be imprecise because they have not been adjusted by instrumental variables.

The sex ratio (*SR*) and indicators about industrial structure including the growth rate of tertiary industry GDP (*GRT*), the growth rate of secondary industry GDP (*GRS*), the proportion of secondary industry in GDP (*PSI*), and the urbanization rate (*UR*) were regressed in Model 4, Model 5, and Model 6. We focused on the relationship between the sex ratio and the growth rate of tertiary industry GDP, while *GRS*, *PSI*, and *UR* were put in models as covariates. The growth rate of RGDP per capita (*GRR*) had no direct relationship with sex ratio (not shown in the table), while the growth rate of RGDP per capita was closely related to the growth rate of tertiary industry GDP (coefficient = 0.855, P < 0.001). Thus, the growth rate of RGDP per capita was used to instrument the growth rate of tertiary industry GDP in the first-stage in Model 5, while the growth rate of RGDP per capita and the growth rate of tertiary industry GDP lagged by one year were used to instrument the growth rate of tertiary industry GDP in Model 6. The *F* value of the first stage in Model 5 was larger than 10, which indicated the growth rate of RGDP per capita was not a weak instrument variable. The 2SLS model (Model 5) found

that a lower sex ratio (higher proportion of female) was associated with a faster growth rate of tertiary industry GDP (coefficient = −1.635, P = 0.033), which was consistent with the results of the OLS model (Model 4) (coefficient = −0.741, P = 0.031) and Model 6 (coefficient = −0.972, P = 0.044), indicating that the negative correlation was robust. This result meant that the sex ratio decreased by 1.63 percentage points for every 1 percentage point of the growth rate of tertiary industry GDP increase. Also, we found a higher growth rate of secondary industry GDP and a higher proportion of secondary industry in GDP were associated with a lower proportion of females in Model 5 and Model 6. Even the coefficients may be imprecise because they have not been adjusted by instrumental variables.

The youth dependency ratio (*YDR*), the proportion of tertiary industry in GDP (*PTI*), and educational investment (*EI*) were cointegrated of the same order. We used FMOLS (Model 7) and DOLS (Model 8) to analyze the relationship between the proportion of tertiary industry in GDP, the educational investment, and the youth dependency ratio. Both Model 5 and Model 6 found provinces with more educational investment tended to have a higher proportion of children among the immigrants (coefficient = 0.012 and 0.014, respectively, P < 0.001). The influences of the proportion of tertiary industry in GDP on the youth dependency ratio were inconsistent in the two models. The results of DOLS were more accurate due to the small samples, which indicated there was no significant correlation between the proportion of tertiary industry in GDP and the youth dependency ratio (coefficient = −0.127, P = 0.095). In Model 9, because of the limitation of sample size (observations = 18), we could only use the OLS method to analyze the relationships between the financial expenditure per capita on education (*FEE*), the financial expenditure per capita on culture and recreation (*FEC*), the financial expenditure per capita on health (*FEH*), and the elderly dependency ratio (*EDR*). The results indicated that provinces with more financial expenditure per capita on culture and recreation (coefficient = 153.31, P = 0.023) had a higher proportion of elderly immigrants.

**Table 6.** Results of panel data models.

| Variables | Dependent Variable: ANM | | | | Dependent Variable: SR | | | | Dependent Variable: YDR | | Dependent Variable: EDR |
|---|---|---|---|---|---|---|---|---|---|---|---|
| | Model 1 (OLS) | Model 2 (2SLS) | | Model 3 (2SLS) | Model 4 (OLS) | Model 5 (2SLS) | | Model 6 (2SLS) | Model 7 (FMOLS) | Model 8 (DOLS) | Model 9 (OLS) |
| | | GRR (First Stage) | ANM (Second Stage) | | | GRT (First Stage) | SR (Second Stage) | | | | |
| GRR# | −6.459 (0.000) | | −7.829 (0.000) | −6.442 (0.000) | | 0.855 (0.000) | | | | | |
| GRT | 3.625 (0.000) | 0.456 (0.000) | 4.643 (0.000) | 4.263 (0.000) | −0.741 (0.031) | | −1.635 (0.033) | −0.972 (0.044) | | | |
| GRS | 3.144 (0.000) | 0.539 (0.000) | 3.885 (0.000) | 2.883 (0.003) | 0.402 (0.026) | −0.383 (0.000) | 0.672 (0.003) | 0.442 (0.016) | | | |
| UR * | | −0.520 (0.000) | | | −0.063 (0.362) | 0.101 (0.042) | −0.104 (0.009) | −0.114 (0.000) | | | |
| PSI | | | | | 0.229 (0.049) | 0.226 (0.000) | 0.188 (0.029) | 0.225 (0.000) | | | |
| PTI | | | | | | | | | −0.134 (0.003) | −0.127 (0.095) | |
| EI | | | | | | | | | 0.012 (0.000) | 0.014 (0.000) | |
| FEE | | | | | | | | | | | −9.806 (0.608) |
| FEC | | | | | | | | | | | 153.313 (0.023) |
| FEH | | | | | | | | | | | −40.006 (0.269) |
| Province dummies | Yes | Yes | Yes | Yes | No | Yes | No | No | – | – | No |
| Year dummies | Yes | Yes | Yes | Yes | Yes | Yes | Yes | Yes | – | – | Yes |
| *F*-value | 26.19 | 74.99 | 13.72 | 11.00 | 6.12 | 11.88 | 11.35 | 10.88 | – | – | 3.19 |
| Overall $R^2$ | 0.61 | 0.97 | 0.87 | 0.89 | 0.33 | 0.71 | 0.48 | 0.70 | 0.79 | 0.79 | 0.49 |

Note: ANM, annual net migration rate; SR, sex ratio; YDR, youth dependency ratio; EDR, elderly dependency ratio. GRR, growth rate of RGDP per capita; GRT, growth rate of tertiary industry GDP; GRS, growth rate of secondary industry GDP; UR, urbanization rate; PSI, the proportion of secondary industry in GDP; PTI, the proportion of tertiary industry in GDP; EI, educational investment; FEE, financial expenditure per capita on education; FEC, financial expenditure per capita on culture and recreation; FEH, financial expenditure per capita on health. Province dummies (Yes) represents the individual intercept is fixed, while the Province dummies (No) represents the individual intercept is random. Year dummies (Yes) represents the individual trend is fixed, while the Year dummies (No) represents the individual trend is random. *, the *UR* was used to instrument for *GRR* in Model 2, and the *UR* and *GRR* lagged by one year were used to instrument for *GRR* in Model 3; #, the *GRR* was used to instrument for *GRT* in Model 5, the *GRR* and *GRT* lagged by one year were used to instrument for *GRT* in Model 6. For the FMOLS estimation method we used heterogeneous long-run coefficients in the first stage of residual calculation. For the DOLS estimation method the optimal number of leads and lags was selected using SIC criterion. *P*-values are reported in parentheses.

## 4. Discussion

Along with rapid urbanization and economic development, China is experiencing the largest internal population migration in the world. This study is the first to describe the trends of the number, geography, age, and gender structure of inter-provincial migrants in mainland China from 2011 to 2016 and analyze the correlation between the number of net inflow migrants, the gender–age structure of inter-provincial migrants, and economic factors in active net-immigration provinces in China. We found that the annual net migration rates were negatively related to the growth rate of RGDP per capita and positively related to the growth rate of industry. The proportion of women among migrants was closely related to the industrial structure and the growth rate of the industry, while the age structure of inter-provincial migrants was related to regional economic investments in education and financial expenditure on culture and recreation. The findings of this study further reveal the law of inter-provincial population mobility, and they provide policy reference for the management of migrants, which can be useful to not only mainland China but also other developing countries that face internal migration.

The result of province types was consistent with the actual distribution of migrants [7]. Most of the net inflow provinces are coastal provinces with relatively developed economies, while Xinjiang having a large number of immigrants might be due to the "National Aid Program for Xinjiang" [70]. While Tibet, which was considered to have a high net inflow rate in a previous study [71], was divided into areas with inactive migration after adjustment of indicators in our study, which is in line with the actual situation of Tibet's population. This result showed that the inter-provincial migrants in China moved mainly from the central provinces to the eastern coastal provinces in 2010, which was consistent with the results of the 2000 census data [8]. In active net-immigration provinces, the proportions of inter-provincial migrants from neighboring provinces obviously increased, and the proportions of migrants from distant provinces declined, especially Chongqing and Sichuan (in western China), which indicated that the growth rate of migration from distant provinces was decreasing compared with that from adjacent provinces. This phenomenon could be explained by the Everett push–pull theory [23]. Because the central government has vigorously implemented the "Western development policy" since 2001 [72], the economic gap between the western provinces and the active net-immigration provinces has narrowed. This has reduced the pull factors of migration, while the distance of migration, one of the push factors, has remained constant. For inter-provincial migrants traveling over a long distance, the pull factors may have become outweighed by the push factors when the economic gap between the origin and destination provinces narrowed, leading them to engage in return migration. For inter-provincial migrants traveling over a short distance, the push factors were constantly weak, so they chose to continue migrating even if the pull factors were weakened. With decreasing inter-regional economic differences, it can be predicted that distance, which was neglected in previous studies focusing on return migration [73], may play a more significant role in future migration.

The annual net migration rate (*ANM*) (mean = 6.1%) varied obviously with time and province. With the decrease of fertility intention in China, inter-provincial migration has become the main motive for population growth in some provinces such as Beijing and Tianjin. Meantime, we found the trends of annual net migration rates were declining, which was confirmed in the annual report of migrants in China [7]. Further, the annual net migration rate was found to be related to economic factors. The results of Model 1, Model 2, and Model 3 in Table 6 indicated a higher annual net migration rate was associated with lower growth rate of RGDP per capita (*GRR*). Previous cross-sectional studies have shown the annual net migration rate was positively related to the GDP per capita of the region [29,41]. While regions with higher GDP per capita were found tend to have a lower growth rate of RGDP per capita [74]. On the one hand, it was difficult to maintain the original growth rate as the economic volume grew. On the other hand, the continuous inflow of a large number of migrants reduced the per capita resources, which might hinder the creation of wealth of people; while the increase of employment competition led to the problem of unemployment [75], which might lead to the decrease of growth rate of GDP per capita. Also, we found that the growth of secondary and tertiary industries could

still attract inter-provincial migrants, and the tertiary industry was more attractive than the secondary industry. Previous research in the United States showed that the tertiary industry could create more economic value than the traditional secondary industry, which led migrants in the United States to move to coastal states instead of to the heavy industrial zones in the western states [3]. Previous data also showed that new-generation migrants were more engaged in the tertiary industry than the elder generation in China [76], which was validated from a macroscopic view in this study. These findings further reveal the relationship between migration and economic factors. As the economy develops, how to provide migrants enough social services and avoid the situation where many internal migrants live in urban villages or slums, which seriously hinders the process of urbanization, as observed in India [5], is a matter of concern.

The sex ratio of inter-provincial migrants (mean = 111.5%) in many provinces was declining, especially in Shanghai, one of the most developed and outward-looking areas in China; this was consistent with the trend of the sex ratio of migrants in many developed countries and regions, such as the United States [13] and the EU [77]. Furthermore, compared to previous studies on the relationship between the economy and female migrants [13,31,77], we included comprehensive economic indicators in Model 4, Model 5, and Model 6 in Table 6, and we found that development of the tertiary industry was related to the flow of inter-provincial female migrants, while the development of secondary industry might not promote female migration in China. In economically developed regions, more employment opportunities suitable for women have been provided in the service industry, while traditional secondary industries, such as construction and manufacturing, have attracted mostly male migrants [13]. Thus, provinces with a higher proportion and faster development of the tertiary industry need to allocate more resources to maternal health.

The youth dependency ratio (*YDR*) of inter-provincial migrants in active net-immigration provinces (mean = 23.9%) was still relatively low compared with that in EU countries [77]. Since migrant children in China find it hard to obtain good education and social welfare in the destination province [42], they were more likely than migrant children in Europe to be left in the original province under the care of their grandparents. In this study, except for that in Guangdong province, the youth dependency ratio in other active net-immigration provinces did not increase over time, which implied that the phenomenon of left-behind children may have not been ameliorated in recent years. The results of Model 7 and Model 8 indicated that the youth dependency ratio was positively related to educational investment. Studies in Sweden and the UK have also found that educational resources are an important factor in students' migration to metropolitan areas [78,79]. In Chinese provinces with more educational investment, inter-provincial migrants tend to take their children with them rather than leave them in the original province as left-behind children [35], which was also validated from a macroscopic view in this study. At present, in the absence of their parents' company and guardianship, not only in China but also in Ukraine [80] and India [81], the inferior physical and mental health and school dropout rates of left-behind children have attracted much attention. Meanwhile, the arrival of migrant children could promote parental settlement intention [82], so as to alleviate the problem of aging population in the province. This finding provided feasible policy recommendations to promote the mobility of children to alleviate these problems.

The elderly dependency ratio (*EDR*) of inter-provincial migrants in active net-immigration provinces (mean = 3.3%) was much lower than that in developed countries, such as 6% in the EU [77]. Compared with the retired elderly in developed countries, who actively migrate to the Sunbelt [15,80], the elderly in China tend to spend their retired life in their hometown. This is because Chinese migration mostly is due to economic reasons [8]. Once migrants can no longer earn money, they lose their motivation to stay in cities [23]. The results of Model 9 showed that elderly inter-provincial migrants fully considered the richness of the cultural and recreational activities in the destination. A previous study similarly found that the number of listed buildings, one of the indicators reflecting the richness of culture [83], was positively correlated with the number of elderly migrants in the UK [25]. Generally, migrants were more often exposed to potentially health-damaging work environments than

were local laborers [84], and the decline of health status with age was more obvious among migrants than among local people. Moreover, the current group of elderly migrants was born around the 1950s. They grew up in poor hygienic conditions and received little health education, so their needs for medical and pension resources are very strong after their long-term migration [39]. This situation may be similar to what many developing countries, such as Vietnam, will face in the future [85]. Therefore, regions with more financial expenditures on culture and recreation might face greater pressures of aging, and how to satisfy the needs of elderly migrants is a matter of concern in these regions.

We have confirmed in our results that instrumental variables were closely related to independent variables and not directly related to dependent variables. Further, we also have to discuss whether instrumental variables are indirectly related to dependent variables through unmeasured variables. Economy, geography, and climate are empirically associated with the annual net migration rate [40], while in Model 2, the urbanization rate is unlikely to change annual net migration rate through geography and climate factors. In Model 5, growth rate of RGDP per capita might promote women's independent migration through the promotion of women's social status [86], thus negatively affecting the sex ratio of inter-provincial migrants. Thus, to the extent that the hypothesized bias exists, our estimates would be in the lower bounds on the true impact of growth rate of tertiary industry GDP on sex ratio. Nonetheless, we acknowledge that we are unable to definitively rule out the possibility that instruments could have some independent impact on the dependent variables beyond its impact working through economy, though we believe that these other effects would not affect the robustness of the results. While in Model 9, because of the limitation of sample size, the relationship between elderly dependency ratio and economic factors has not been explored in depth, and the robustness of the results require further study.

Our study had some limitations. First, in China, the emergence of internal migrants is linked to the household registration system. However, after more than 60 years of practice and reform, the household registration policy differs in each province, so it was difficult to quantify the impact of the policy on the number and structure of the inter-provincial migrants in this study. Second, because of the lack of data concerning the outflow migrants of provinces, this study did not analyze the demographic characteristics of the migrants leaving the active net-emigration provinces or any relationships with economic factors. Such information could help us better understand how economic factors affect population mobility and support decision-making regarding regional development and health. Third, some confounding factors related to migration, such as population density [87], were not included in this study.

## 5. Conclusions

Our study is the first to present findings suggesting that the number and gender–age structure of inter-provincial migrants in China are closely related to economic indicators. As the economy develops, these results provide insights for migration policy-making and regional development planning considering the changes in the number and gender–age structure of the migrants. A younger demographic structure of the population is crucial to sustainable urban development. Our findings suggest that for areas facing the aging of population, promoting the development of the tertiary industry and increasing education investment might be effective measures to promote household migration and settlement of migrants, so as to alleviate the problem caused by an aging population and left-behind children. Additionally, compared with the secondary industry, work opportunities in the tertiary industry could be more attractive to migrants, especially females. Along with the fast development in the tertiary industry, it could be necessary and important to allocate maternal health resources in areas with a high proportion of female migrants to meet their health needs. Reasonable management of the migrants could be conducive to improving their living environment and help them to devote to work, and further provide more power for the sustainable development of the city.

As a country with the largest internal migrants in the world, China has experienced fast urbanization processes in the past three decades. The findings about migration and economy in

China have an important reference value for other countries who are facing internal migration and are experiencing the process of urbanization. In order to further verify the correctness of these findings, we will further test the prediction ability of these economic factors on the size and structure of inter-provincial migrants in future studies.

**Author Contributions:** Conceptualization, L.L. and W.C.; methodology, L.S.; formal analysis, L.S.; data curation, L.L. and J.X.; writing—original draft preparation, L.S.; writing—review and editing, L.S. and W.C. All authors have read and agree to the published version of the manuscript.

**Funding:** This work was supported by the China Medical Board grant number (12-111).

**Acknowledgments:** All authors thank the editor and the anonymous reviewers for their constructive comments and suggestions for improving the quality of this paper. All authors thank the Migrant Population Service Center of the National Health Commission of the People's Republic of China for providing the data.

**Conflicts of Interest:** The authors declare no conflicts of interest.

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
