# Peer review of "Trends and Characteristics of Inter-Provincial Migrants in Mainland China and Its Relation with Economic Factors: A Panel Data Analysis from 2011 to 2016"

_sustainability, doi:10.3390/su12020610_

Round 1

Reviewer 1 Report

This paper aims at descriptions "of the changing trend of inter-provincial migrants in China from 2011 to 2016" and at "analysing the dynamic relationship between economic factors and the number and gender-age structure of inter-provincial migrants". The descriptive part of the paper and the literature review seems to be relatively well prepared. The authors have updated all the sections according to the previous comments. The authors improved especially the part related to the panel data model regression.

My comments focus on the econometric part of the paper. Although the authors estimated the model by the 2SLS method, they do not discuss the validity of the instruments. The author should address the problem of endogeneity of the instruments used. Including more than one instrument (e.g. the lagged explanatory variable) allows testing their validity formally. Why was the Hausman-Taylor IV estimator not used for the panel data model?

Using the short names of the variables in the discussion section is confusing. Please, use the full names of the variables (or their meaning), the short names may be mentioned in the parentheses. Using the estimated values of the coefficients and p-values in the main text (discussion) is a little disturbing (one may find these values in the corresponding table).

The concluding section is too short, and it can be merged with the Discussion section.

Author Response

This paper aims at descriptions "of the changing trend of inter-provincial migrants in China from 2011 to 2016" and at "analysing the dynamic relationship between economic factors and the number and gender-age structure of inter-provincial migrants". The descriptive part of the paper and the literature review seems to be relatively well prepared. The authors have updated all the sections according to the previous comments. The authors improved especially the part related to the panel data model regression.

My comments focus on the econometric part of the paper. Although the authors estimated the model by the 2SLS method, they do not discuss the validity of the instruments. The author should address the problem of endogeneity of the instruments used. Including more than one instrument (e.g. the lagged explanatory variable) allows testing their validity formally. Why was the Hausman-Taylor IV estimator not used for the panel data model?

Author’s reply: We thank the reviewer for pointing out this problem. In the result section (rows 392-398 and rows 409-413) and discussion section (rows 554-568), we discussed the validity and exogenous of instrumental variables, including whether the instrumental variables may affect dependent variables through variables not measured in this study. We also followed the reviewer's opinion and used multiple instrumental variables (including the lagged explanatory variables) to analyze the models. The results showed that the correlations between variables were robust (See Table 6). Compared with normal instrumental variable estimator, the Hausman-Taylor IV estimator has the advantage of being able to estimate the influence of factors that do not change with time (such as distance between regions, policies on migration, etc.) on the dependent variables. However, since all the independent variables in this study were time-varying variables, the normal instrumental variable estimator was adopted.

Using the short names of the variables in the discussion section is confusing. Please, use the full names of the variables (or their meaning), the short names may be mentioned in the parentheses. Using the estimated values of the coefficients and p-values in the main text (discussion) is a little disturbing (one may find these values in the corresponding table).

Author’s reply: We thank the reviewer for pointing out this problem. We have revised all content in discussion section as the reviewer’s suggestion.

The concluding section is too short, and it can be merged with the Discussion section.

Author’s reply: We thank the reviewer for pointing out this problem. We have enriched the content of the conclusion section, and made a comprehensive summary of the results and discussion (See rows 580-599).

Reviewer 2 Report

This paper explores urbanization processes in mainland China associated with migration and demographic change. The paper should appeal to readers of the journal Sustainability. I recommend the following changes, and then publication.

The authors should further engage and synthesize the literature on the urbanization of China. What are the driving factors associated with these changes? The authors should consider, for example, the work of Ren (2013) on "Urban China."

The authors should expand the conclusion to fully develop it. What are the implications of the research? How might planners and policymakers respond? What are the future research prospects? How can we learn from the example of China? What are implications and relationship to sustainability?

Thank you.

Author Response

This paper explores urbanization processes in mainland China associated with migration and demographic change. The paper should appeal to readers of the journal Sustainability. I recommend the following changes, and then publication.

The authors should further engage and synthesize the literature on the urbanization of China. What are the driving factors associated with these changes? The authors should consider, for example, the work of Ren (2013) on "Urban China."

Author’s reply: We thank the reviewer for pointing out the shortcomings of the literature review. Correspondingly, we have added the content about urbanization and population mobility in China as the background of the current study (See rows 56-65).

The authors should expand the conclusion to fully develop it. What are the implications of the research? How might planners and policymakers respond? What are the future research prospects? How can we learn from the example of China? What are implications and relationship to sustainability?

Author’s reply: We thank the reviewer for pointing out this problem. We have enriched the content of the conclusion section, and made a comprehensive summary of the results and discussion. A younger demographic structure of the population is crucial to the sustainable urban development, our findings suggest that for areas facing the aging of population, promoting the development of the tertiary industry and increasing education investment might be effective measures to promote the household migration and settlement of migrants, so as to alleviate the problem cause by aging of population and left behind children. Additionally, compared with the secondary industry, work opportunities in the tertiary industry could be more attractive to migrants, especially females. Along with the fast development in the tertiary industry, it could be necessary and important to allocate maternal health resources in areas with high proportion of female migrants to meet their health needs. Reasonable management of the migrants could be conducive to improving their living environment and help them to devote to work, and further provide more power for the sustainable development of the city. As a country with the largest internal migrants in the world, China has experienced fast urbanization process in the past three decades. The findings about migration and economy in China have important reference value for other countries who are facing the internal migrants and experiencing the process of urbanization. In order to further verify the correctness of these findings, we will further test the prediction ability of these economic factors on the size and structure of inter-provincial migrants in future studies. (See rows 580-599).

Round 2

Reviewer 1 Report

The authors accepted most of my previous comments. I have thus only minor comments:

row 421 "YDR, PTI and EI were cointegrated of the same order" instead of "were cointegration..." the authors could improve the readability of the text by using full names of the variables in section 3

Author Response

The authors accepted most of my previous comments. I have thus only minor comments:

row 421 "YDR, PTI and EI were cointegrated of the same order" instead of "were cointegration..." the authors could improve the readability of the text by using full names of the variables in section 3.

Author’s reply: We thank the reviewer for pointing out these problems. We have modified the corresponding statement (See row 439) and used the full name of all variables in the text in section 3.

This manuscript is a resubmission of an earlier submission. The following is a list of the peer review reports and author responses from that submission.

Round 1

Reviewer 1 Report

This paper aims at descriptions "of the changing trend of inter-provincial migrants in China from 2011 to 2016" and at "analysing the dynamic relationship between economic factors and the number and gender-age structure of inter-provincial migrants". The descriptive part of the paper seems to be relatively well prepared. On the other hand, the value-added of the panel model regression is questionable.

The abstract of the submitted paper is not entirely following its content. One can agree with the statement that the study analysed (described) the trends and characteristics of inter-provincial migration flows. However, there is a lack of an in-depth analysis of the possible causal relationship of this phenomenon with possibly related economic factors. The authors estimated panel data regression, but they omitted the problem of endogeneity of the analysed variables. The authors try to explain the correlation between the migration flows (and socio-demographic characteristics of these flows) and the economic development (measured by using various indicators of economic growth or social capital investment). The issue of the influence of migration flows on economic growth is neglected.

In the abstract, the authors state that "this study provides a theoretical basis for forecasting and adjusting the trends" of migration. The similar statement is occurring in the concluding section. Unfortunately, the proposed model does not allow us to predict or forecast migration patterns- To fulfil this task, one has to formulate the model by incorporating lagged explaining variables. One has to test its prediction abilities as well.

The coefficients of determination are hard to interpret in panel data regression models (moreover, the authors should clarify which one they have computed: overall R-squared, within- or between R-squared).

The literature review is well prepared. The authors describe the models for studying migration patterns. How is the model presented in the paper related to these models? The literature shows that "the relationship between migration and economic development is bidirectional". The presented paper does not discuss this crucial issue!

It is hard to believe that no study has discussed the covered topic in China. What about Gries et al. (2015) - https://doi.org/10.1111/pirs.12156, Su et al. (2017) - http://ftp.iza.org/dp11029.pdf, or Qi et al. (2017) - doi.org/10.1177/0308518X17718375?

The presented study uses modified indicators of the migration rates without discussing its advantages to the primary indicators and without the robustness check of the obtained results concerning the indicators used.

The authors use panel unit root tests and the Pedroni panel cointegration test. It is not clear how the using of these test may avoid "pseudo regression" (I guess that the author meant spurious regression). The description of the empirical methodology (as well as some presented results) should be improved and rewritten following standard econometric terminology (e.g. single difference = first difference). The Hausman test is intended to test the appropriateness of the random effect model (i.e. not its preference to the fixed-effect model). The results in Table 6 indicate the p-value of the Hausman test at the value of 1. Why was thus the random-effect model not used, at least for robustness check?

Figure 2 is missing (see row 267). The Table 6 could be better arranged (the estimates for particular models could be presented in separate columns, and the first column could contain variables used in all models).

The author should discuss the problem of endogeneity and causality among the analysed variables.

Reviewer 2 Report

The main topic of this paper, inter-provincial migration in China and their correlation with some macro-economic data, is of considerable interest.

The most interesting part of this paper consists of... raw data on inter-provincial migration (table 2 and table 3), but also these data are poorly presented, e.g., their description is (in fact) in section 2.1, where there is no explicit reference to the tables.  It is a pity that analogous raw data for economic variables are not presented at all (at least in the form of a supplementary material). 

The statistical results are presented very briefly (just two pages 10-11), which is completely unacceptable taking into account that the Authors claim to test four different models and many statistical tools and tests. As a minimum some information about the meaning and/or significance of the statistical output is needed. Now, it looks like a kind of computer output without any discussion and any differentiation between meaningful and neutral results. The Authors assume linear dependence of data (typical in economical sciences, but often not very useful - many dependencies can be nonlinear, linearity is rare) and put it into a computer. The discussion in section 4 is surprisingly unrelated to results of section 3 (just in few places, without reference to any numbers, so abundant in section 3).

More detailed remarks and criticism:

Lines 129 and 130 (and many other places): please use the same font wherever a given notation appears. e.g., RNMi is in italics in line 129 and in normal font in line 130.

The second part of the formula (3) is probably wrong (because it yields RMNi negative), perhaps modulus (absolute value) is needed.

The formula (4) is wrong. Maybe a plus is missing?

Both formulas, (3) and (4) are not motivated and the indicators defined by these formulas are, in my opinion, rather useless. 

Tables 1 and 2 cannot be compared because different indicators are used (NM is the table 2, and this strange RMN in the table 1). I would suggest to resign from using RMN altogether, or to explain its meaning (e.g., what does it mean - intuitively - if it exceeds 100%). 

The formula (5), crucial to obtaining the important coefficient Ra, has to be completed with a definition of "national average gross migration rate for all provinces". 

The coefficient Ra seems to be defined in a strange way in the formula (5) because it is a product of two rates. Please explain and/or motivate. Table 1: why the net migrants (column 6) do not sum up to zero?  

Lines 267-268:  Figure 2 is missing. 

Lines 113-118 and Table 4: why RGDP per capita is not used? It seems to be one of the best indicators of pull factors... 

It is quite probable that the data handled by the Authors can be used to generate interesting results and a good paper. However, it needs (in my opinion) much more time than just few days. The first point to be done is to extract some interesting dependencies from the vast volume of the raw data (this is hardly done in the submitted paper) and then, to present these dependencies in a clear and convincing way (this is not done at all).  Therefore, I recommend to reject the paper in its present form. Perhaps it may be resubmitted later, after a really major revisions.

Reviewer 3 Report

The topic is interesting but this paper has nothing to do with sustainability and sustainable development (or only extremely loosely). It should be submitted to a migration journal (I would suggest Asian and Pacific Migration Journal or Migration, Mobility & Displacement) or to a generic geography journal (I would suggest (for ex.) Professional Geographer or Geographical analysis or Asian Geographer).